# Chitosan-Based Nanoparticles for Cardanol-Sustained Delivery System

**DOI:** 10.3390/polym14214695

**Published:** 2022-11-03

**Authors:** Roberta Bussons Rodrigues Valério, Nilvan Alves da Silva, José Ribamar Paiva Junior, Anderson Valério Chaves, Bruno Peixoto de Oliveira, Nágila Freitas Souza, Selene Maia de Morais, José Cleiton Sousa dos Santos, Flávia Oliveira Monteiro da Silva Abreu

**Affiliations:** 1Programa de Pós-Graduação em Ciências Naturais, Universidade Estadual do Ceará, Fortaleza 60714-903, CE, Brazil; 2Departamento de Química Analítica e Físico-Química, Campus do Pici, Universidade Federal do Ceará, Fortaleza 60455-760, CE, Brazil; 3Instituto de Engenharias e Desenvolvimento Sustentável, Universidade da Integração Internacional da Lusofonia Afro-Brasileira, Campus das Auroras, Redenção 62790-970, CE, Brazil

**Keywords:** chitosan, cardanol, encapsulation, nanomaterials, antioxidants

## Abstract

Cardanol, principal constituent of the technical cashew nut shell liquid, has applications as antioxidant and antibacterial, and these properties may be enhanced through encapsulation. In the present study, we isolated and purified cardanol, and nanoparticles (NPs) were produced by polyelectrolyte complexation using polysaccharide systems with chitosan, sodium alginate, and non-toxic Arabic gum, because they are biocompatible, biodegradable, and stable. We characterized the NPs for morphological, physicochemical, and antioxidant activity. The micrographs obtained revealed spherical and nanometric morphology, with 70% of the distribution ranging from 34 to 300 nm, presenting a bimodal distribution. The study of the spectra in the infrared region suggested the existence of physicochemical interactions and cross-links between the biopolymers involved in the encapsulated NPs. Furthermore, the NPs showed better antioxidant potential when compared to pure cardanol. Thus, the encapsulation of cardanol may be an effective method to maintain its properties, promote better protection of the active ingredient, minimize side effects, and can target its activities in specific locations, by inhibiting free radicals in various sectors such as pharmaceutical, nutraceutical, and biomedical.

## 1. Introduction

Cardanol is a byproduct of the cashew processing industry, and its global production was estimated to be 4.1 million tons in 2016, according to the Food and Agriculture Organization. Cardanol is the main constituent of the technical cashew nutshell liquid (tCNSL) and has potent antioxidant activity [1,2,3] because it is made up of a mix of aromatic and phenolic compounds. Thus, cardanol is one of the most essential green industrial raw materials and an abundant, refined, promising source of renewable phenols [4] because it possesses a critical group that eliminates free radicals formed during the oxidative process [5].

In recent years, there has been an increasing interest in studies on the evaluation and development of antioxidants from biomass, industrial and renewable waste, due to environmental concerns, limited availability of fossil resources, and the inconvenience of thermo-oxidative degradation of various organic products and processes [6,7,8,9]. Aiming to develop cheaper and more eco-friendly antioxidants with appropriate physicochemical properties [1,2,10,11], polysaccharide-based nanoparticles (NPs), chitosan (CHI) [12,13], sodium alginate (ALG), and Arabic gum (AG) have been used to encapsulate active ingredients by polyelectrolyte complexation (CPE) in their hydrophilic polymer matrices. This method uses ionic interactions between compounds that integrate opposite charges and utilizes routes that avoid the use of organic solvents [14,15]. They are colloidal structures that coat drugs, aiming to avoid the degradation of the active ingredient, improve its efficacy, circumvent its toxicity and considerably prolong its biological activity [16].

The present work aimed to isolate cardanol from (tCNSL) and produce core-shell polysaccharide NPs by using polyelectrolyte complexation. The NPs were subjected to physicochemical characterization, and morphology, antioxidant activity, and total phenol content were observed. NPs of the CHI complex with ALG or AG were produced, with significant advantages regarding drug stability. This enabled protection of the active ingredient and its prolonged release.

## 2. Materials and Methods

### 2.1. Materials

The chitosan (CHI) polymer we used in the present work originates from crabs and was obtained, purified, and characterized as described in Abreu et al. (2013), with a viscosimetric molar mass (Mz) of 4.126 × 105 g mol^−1^ and a deacetylation degree of 77% [17]. Sodium alginate (ALG, Sigma-Aldrich, St. Louis, MO, USA) and Arabic gum (AG, Dinâmica, SP, Brasil) sodium tripolyphosphate (TPP), and Tween^®^ 80 (Dinâmica Química Contemporânea, Jardim da Glória, São Paulo, SP, Brasil) were also used. Cardanol was isolated and purified according to a methodology with adaptations [18], monitored by thin-layer chromatography (TLC), and analyzed by HPLC (high-performance liquid chromatography).

### 2.2. Nanoparticles Synthesis

The NPs were synthesized according to the following methodology [15]. Initially, the cross-linking stage with subsequent formation of the polyelectrolyte complex was prepared. A solution containing 50 mL of 1% CHI and 100 µL of Tween^®^ 80 were mixed under mechanical stirring. Subsequently, cardanol was added in a 10:1:1 volumetric ratio to the solution and submitted to the ultrasonic bath (Ultra 800, Ciencor Scientific Ltda, São Paulo, Brazil) for 15 min. Then, 2 mL of the TPP solution were added in a drip at concentrations of 0.1 and 0.01 mol L^−1^, to form a kind of pre-core for the NPs and the solution was dispersed under magnetic stirring for 30 min. After that, 1 mL of the ALG solution was added dropwise to the solution and mixed for another 30 min, coating the inner core with ALG chains, forming CHI/TPP/ALG NPs. It was also prepared CHI/TPP/AG NPs, where it was used arabic gum as an external coating instead of ALG. Finally, all prepared NPs were centrifuged at 10,000 RPM (Novatécnica NT 810) and dried by lyophilization (L101, Liobras^®^, Carlos, Brazil).

### 2.3. Characterization of Cardanol-Loaded NPs

Physicochemical characterization was performed for NPs loaded with optimal cardanol formulations regarding yield and encapsulation efficiency, chosen through ANOVA statistical analysis in a previous study [19]. The cardanol-loaded NPs were characterized by FT-IR on a Shimadzu FTIR 8300 device (Shimadzu, Kyoto, Japan) in the 4000 to 400 cm^−1^ range.

The morphology and aggregation profile of the NPs were evaluated by scanning electron microscopy (SEM). The samples were fixed on metal supports, covered with carbon tape, and coated with gold for electrical conductivity generation by means of a metallizer Quorum Metallizer QT150ES (Quorum Technologies Ltd., East Sussex, UK). A 20 nm layer of material was formed. Micrographs were obtained on a scanning electron microscope Quanta FEG 450 Electron Microscope, FEI Environmental (Eindhoven, The Netherlands) with an accelerating voltage of up to 20 kV.

The size distribution and surface measurement of zeta potential-charged NPs were determined by DLS (dynamic light scattering) by means of a Nano Zeta Sizer Malvern 3600 (Malvern Instruments Ltd., Malvern, UK). Aliquots of the 1 mL sample were taken and diluted in 1:100 *v*/*v* deionized water. The analyses were performed under neutral pH conditions at room temperature. Thermogravimetric analysis (TGA) was performed on a TGA/SDTA 851 (METTLER TO-LEDO, Columbus, OH, USA), with ultrapure nitrogen as purge gas at a heating rate of 10 °C min^−1^ with a flow rate of 20 mL/min in an aluminum pan.

### 2.4. In Vitro Release

In vitro drug-release kinetics in NPs was monitored by UV-visible spectrophotometry. A total of 80 mg of cardanol-containing NPs (QUI/TPS/ALG) were added to an analytical filter paper and immersed in 100 mL of phosphate buffer solution at pH 7.0. The release system was kept under magnetic stirring and at a constant temperature of 25 °C. Aliquots of 2.0 mL were withdrawn from the system at determined time intervals and subsequently filtered. Absorbance measurements were performed for all aliquots in a spectrophotometer (Thermo Scientific—Genesys 102S, Waltham, MA, USA) to obtain the total concentration of the released drug.

### 2.5. Determination of Antioxidant Activity

In the evaluation of antioxidant activity, the 2,2-diphenyl-1-picrylhydrazyl (DPPH) free radical scavenging method was used, monitoring the DPPH free radical consumption of the samples by measuring the decrease in UV absorbance. The samples (0.1 mL of methanol solutions) at concentrations ranging from 10,000 to 1 ppm were mixed with 3.9 mL of 6.5 × 10^−5^ M DPPH in methanol, and the UV absorbance of the reaction mixture was analyzed at 515 nm after 60 min [20]. The median inhibitory concentration (IC50) was calculated in an Excel program.

## 3. Results and Discussion

Cardanol, which was isolated by solvent extraction from tCNSL, appeared as expected. Figure 1 shows the HPLC chromatogram obtained.

A relative percentage yield of 87% of the compounds present in the analyzed cardanol was observed in these peaks. Similar results were found by Trevisan et al. (2006) [3]. Tyman and Kiong (1978) were the first to study the separation of CNSL compounds, obtaining 25% cardanol and traces of cardol and 2-methylcardol [21]. Kumar (2002) obtained a significant yield of cardanol (45%) using a single liquid–liquid extraction [22], which is an efficient method for the separation of technical CNSL components. Mazzetto and Lomonaco (2009) obtained 70% pure cardanol using a mixed column of silica gel and celite, with a gradual variation of the eluent [23]. Attanasi (2003) obtained significant yields of cardanol (70–80%) [24].

From the extracted cardanol, core-shell NPs based on chitosan, Arabic gum, and sodium alginate were produced, using a neutral surfactant to provide better solubilization of cardanol oil in an aqueous medium under different reaction conditions. The preparation of the NPs was performed in two distinct steps: first, a central core was formed with CHI and TPP, which acted as an ionotropic crosslinker to ensure the formation of a pre-core by intertwining the chains of the phosphate groups of TPP and the amino groups of CHI.

The CHI cross-linking reaction mechanism and the pre-core formation of CHI-P_3_O_10_^5−^ based NPs are aimed at an increase in acid resistance and chelation efficiency, as well as an increase in drug immobilization capacity [25].

After cross-linking the CHI core, it was subsequently coated with ALG or AG through the interaction between the carboxyl groups of ALG or AG with the CHI amine groups, forming the CHI-ALG and CHI-AG systems with a three-dimensional matrix through the mechanism of polyelectrolyte complexation (PC), using strong ionic electro-static forces as well as intermolecular forces [26].

Equations (1) and (2) represent the chemical reactions of the formation of the CHI-ALG and CHI-AG complexes, respectively.
CHI − NH_3_^+^ + ALG − COO^−^ ↔ CHI − NH_3_^+ −^OOC − ALG(1)
CHI − NH_3_^+^ + AG − COO^−^ ↔ CHI − NH_3_^+ −^OOC − AG(2)

A number of studies have been published using PC to prepare polysaccharide-based NPs, with numerous applications. Tan et al. (2016) synthesized NPs based on chitosan and Arabic gum to encapsulate the antioxidant curcumin [27]. Filho et al. (2019) produced core-shell nanoparticles of alginate and chitosan loaded with anacardic acid and cardol for drug delivery [28].

The properties of NPs depend on the electrostatic attractions between the polymers and the cross-linking agent, as well as the order of their addition in the system. This, in turn, influences the interaction with cardanol [19], which ensures a more excellent protection of cardanol against external environmental factors [28].

A previous experimental work was performed to evaluate the percentage yield and encapsulation efficiency (EE) of the NPs synthesis reactions. In this study, it was proved that using CHI polymer matrix, TPP was the best crosslinker, for the EE of cardanol oil in comparison with ammonium persulfate crosslinker [19]. Figure 2 shows the hypothetical representation of the core-shell NPs particle encapsulation system.

### 3.1. Structural and Morphological Characterization of NPs and Cardanol

In a previous study by the group, cardanol NPs systems, named CHI/TPP/ALG and CHI/TPP/AG, showed the best balance of properties, with values of 50 and 60% for EE and reaction yields of 63 and 48%, respectively [19]. These NPs (CHI/TPP/ALG and CHI/TPP/AG) were produced on a larger scale for further characterization. Characterizing these systems with simplified methodologies is relevant to optimize their efficacy, promote better protection of the active ingredient, and target their action at specific sites, so minimizing side effects [29].

Figure 3 shows the spectra of cardanol, chitosan and encapsulated nanoparticles. In the CHI spectrum, the prominent bands present in the structure of this polymer were evidenced: 3400 cm^−1^ referring to OH/NH stretching, 1580 cm^−1^ [30,31] assigned to NH vibrations, 2916 cm^−1^ referring to the vibrations of C─H bonds, 1654 cm^−1^ assigned to the carbonyl absorption of -NC=O groups of amide II, 1151 cm^−1^ and 1021 cm^−1^ of CO bonds [14]. The spectrum of cardanol showed the following vibrations: OH phenolic stretching (3400 cm^−1^), CH extretching of the aromatic and chain double bonds (3009 cm^−1^), methyl, methylene, and methyl groups (2916 cm^−1^), C=C in the aromatic ring (1609, 1588 and 1505 cm^−1^), symmetric and asymmetric C=C bending (1264 cm^−1^, 1149 cm^−1^), and vibrations of the four hydrogen atoms adjacent to the benzene ring (778 cm^−1^, 691 cm^−1^) [32,33,34]. In the spectrum of sodium alginate, absorption peaks were observed around 2900 cm^−1^ and 1415 cm^−1^, due to the -CH_2_ stretching. A broadband was observed near 3400 cm^−1^, resulting from the axial deformation of OH. It was observed that the deformation of the -COO- ion gives rise to two bands—one at 1415 cm^−1^ and another of higher intensity at 1610 cm^−1^, resulting from symmetric and asymmetric axial deformations, respectively [35,36,37]. It was also possible to observe a band at 1034 cm^−1^, due to symmetric C-O-C axial deformation [35,36,37]. All NPs showed bands at 3400 cm^−1^ referring to OH deformation, and 2924–2927 cm^−1^ and 2852–2854 cm^−1^ attributed to symmetric and asymmetric deformations of the CH_2_/CH_3_ groups present in the cardanol structure [28,29,30,31,32]. In a previous study carried out by the group [28], the increase in the intensity of the band at 3400 cm^−1^ and 1625 cm^−1^, is significantly higher than that of the matrix, suggesting the existence formation of hydrogen bonds between the biopolymers involved in the encapsulated nanoparticles [38,39].

SEM evaluated the morphology of the NPs, and micrographs of the CHI/TPP/ALG, and CHI/TPP/AG NPs are shown in Figure 4. Both revealed a profile of less than 200 nm in size, a rough surface, and spherical particles, with little agglomeration, probably due to the freeze-drying process and variety of fillers present on their surfaces [28,40]. Other polymer systems have also exhibited this phenomenon. The variation in PC reaction conditions can lead to the formation of NPs with different morphology, which can show greater regularity in their spherical structure and particle size [28].

The results for particle size and zeta potential were evaluated as a function of the total volume fraction of NPs loaded with cardanol, shown in Table 1.

We observed that the produced NPs showed a mostly bimodal distribution, with two peaks representing a fraction of particles with different average sizes. For interpretation and analysis purposes, Table 1 shows the average values of each peak for the of NPs CHI/TPP/ALG and CHI/TPP/AG loaded with cardanol.

Similar values to those produced in this study for particle size (34–312 nm) using biodegradable matrices for oil encapsulation have been reported by other authors in the literature. Abreu et al. (2012) encapsulated Lippia sidoides essential oil (between 335 and 551 nm) based on chitosan and cashew gum [41]. Dubey; J. Bajpai and A.K. Bajpai (2016), produced chitosan particles for the adsorption of Hg(II) ions with an average hydrodynamic dimension of 303 and 461 nm [42]. Filho et al. (2019), when working with polysaccharide nanoparticles based on chitosan, alginate, and Arabic gum for encapsulation of anacardic acid and cardol, obtained most nanostructures in both systems, a high fraction of particles smaller than 250 nm [28]. Weibson et al. (2020) produced chitosan nanoparticles loaded with carvacrol and carvacryl acetate with particle sizes of 479, 117, and 204 nm [42].

Zeta potential was investigated for CHI/TPP/ALG and CHI/TPP/AG NPs in aqueous suspension at neutral pH (Table 1); values of −39.7 ± 4.9 mV and −29.8 ± 4.0 mV were observed, respectively. CHI/TPP/ALG NPs showed a higher negative-charge density than CHI/TPP/AG, being comparatively more stable and less predisposed to agglomerate, as observed in SEM micrographs. Particles with zeta potential values >−30 or ≤+30 mV avoid undesirable oscillations, since high zeta potential values provide stability by promoting resistance to aggregation and maintaining colloidal dispersion [14,43]. These negative zeta-potential results indicate that CHI/TPP/AG and CHI/TPP/ALG NPs showed stability, where AG and ALG, anionic polysaccharides, were able to coat the particle surface, formed by chitosan in the inner core [14,28,44].

### 3.2. Thermal Analysis

Figure 5 and Table 2 show the TGA/DTG results for the nanoparticles and cardanol, demonstrating the effect of the chemical treatments on the thermal stability of the samples. Three main thermal degradation events were observed for cardanol and four events for the nanoparticles analyzed.

The first in all samples, 30–150 °C, refers to the evaporation of volatiles, while the others are associated with the decomposition of the main components present in the samples [45,46].

Cardanol showed a second event between 150 and 375 °C, with maximum temperature at 321.3 which may be associated with degradation by oxidation and rupture of the phenolic hydroxyl and aliphatic chains [47]. Furthermore, cardanol presented a third event with maximum temperatures at 475.8 °C, which indicates good thermal stability of this isolated molecule. Cardoso et al. (2018), when encapsulating cardanol using polylactic acid (PLA) microparticles, observed that the chemical concentration of the cardanol additive influenced the thermal stability of the encapsulate, identifying a maximum temperature event of 273 °C in cardanol alone [48]. We also observed that as cardanol was added to the encapsulating up to a specific concentration (50 mg), which fixed the PVA concentration, a second event appeared at 291 °C, indicating an increase in the thermal stability of the encapsulated.

Cardanol, besides having a high molar mass with conjugated carbon chains, has in its constitution one carboxylic group per molecule. This possibly justifies a more significant intermolecular interaction by hydrogen bonding, as well as presenting a mixture of phenols, configuring a more excellent thermal stability against the third event of the nanoparticles analyzed [19,45]. This is corroborated by the FTIR analysis, where it was possible to verify the characteristic bands of the phenolic hydroxyl group.

For encapsulated cardanol samples (CHI/TPP/ALG and CHI/TPP/AG), we observed a second event with a maximum thermal-degradation temperature at 264 °C and a mass loss of 27 and 24%, respectively. This second event is associated with the onset of mass loss of the chitosan polymer. In addition, we can also infer the second event present in isolated cardanol [47]. A third event, with mass losses of 16.5% (CHI/TPP/ALG) and 14.6% (CHI/TPP/AG), with a maximum temperature of 341 °C (CHI/TPP/ALG) and 349 °C (CHI/TPP/AG), for the mentioned samples, suggests the continuation of chitosan degradation [49].

A fourth event (with a slight mass loss in the CHI/TPP/ALG and CHI/TPP/AG samples at 580 °C) refers to the extension of the degradation of the glycosidic ring of the polysaccharide and the degradation of the inorganic components present in the structure of alginate and Arabic gum, configuring a more excellent thermal stability against cardanol [50,51,52].

### 3.3. Release Curve

As mentioned earlier, the main purpose of the microencapsulation process is to encapsulate bioactive compounds into matrices, to protect the compound from the external environment. However, it is also necessary to evaluate how the bioactive compound will be released after its encapsulation [53].

The in vitro test results of the CHI/TPP/ALG NPs demonstrated a small fraction of cardanol initially released, followed by a controlled release rate. According to official compendia, for a formulation to be considered an immediate release act, 78.75% of the amount under consideration must dissolve 30 min [53]. The effect of increasing the degree of cross-linking on drug release in CHI/TPP/ALG NPs is shown in Figure 6. Increasing the degree of cross-linking causes slower drug release.

### 3.4. Evaluation of Antioxidant Activity

The antioxidant activity results, evaluated by the DPPH radical, showed activity with IC 50 in CHI/TPP/ALG (0.470 + 0.021 mg mL^−1^), and CHI/TPP/AG nanoparticles (0.418 + 0.028 mg mL^−1^) showed better antioxidant potential than that of pure cardanol (0.551 + 0.020 mg mL^−1^). The synthesis of new cardanol derivatives is key to the development of industrial applications, adding value to technical LCC. Furthermore, in recent years, the antioxidant activity of cardanol derivatives has been reported to be comparable to commercial products, in this case 2,6-di-tert-butyl-4-methylphenol (BHT) and 2,6-di-tert-butyl-4-methoxyphenol (BHA) [2]. Such advances are considered essential in the antioxidant sector, since the presence of the phenolic structure favors stabilizing applications and allows multiple functionalizations on the hydroxyl.

## 4. Conclusions

The synthesis of NPs based on cardanol-loaded natural polymers was successfully performed, with an average size of less than 200 nm, and expressive antioxidant activity. These properties provide an attractive alternative for replacing synthetic products. The NPs produced are renewable, biodegradable, and produced with low-cost materials from northeast Brazil. Moreover, as an encapsulation protected from degradation, they are a good system for targeting antioxidant applications.

## Figures and Tables

**Figure 1 polymers-14-04695-f001:**
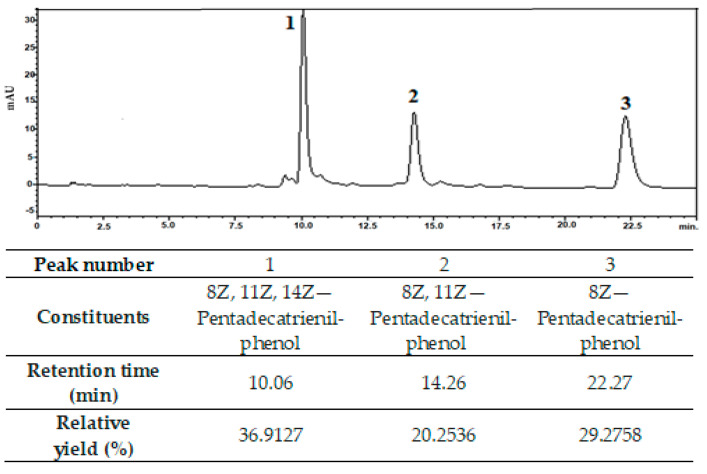
Representative high-performance liquid chromatography profile of cardanol. Column: Hypersil GOLD^®^ 25 cm, run time: 25 min, flow rate: 1.80 L/min, mobile phase: 80:20 acetonitrile:acetic acid (1%).

**Figure 2 polymers-14-04695-f002:**
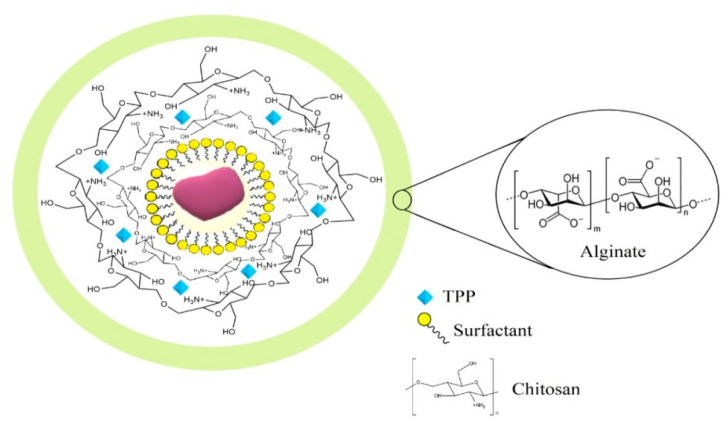
Hypothetical representation of the structure of cardanol-loaded CHI-TPP-ALG NPs.

**Figure 3 polymers-14-04695-f003:**
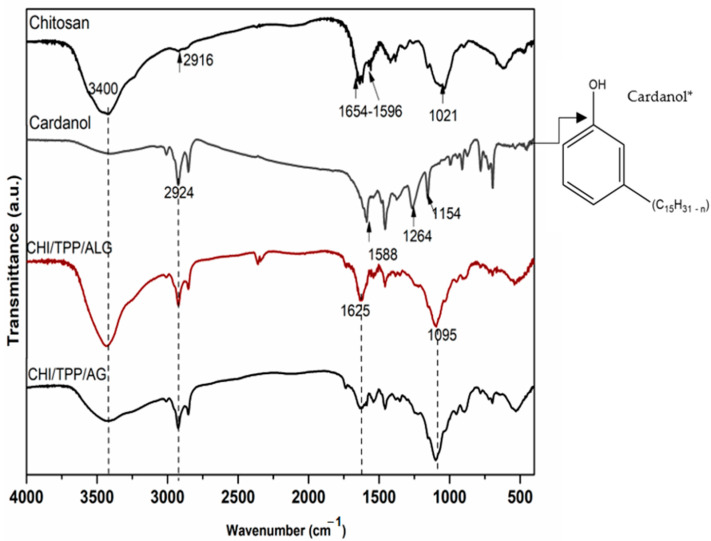
Spectra were obtained in the infrared region for samples of Chitosan, Cardanol* (CHI/TPP/ALG, and CHI/TP/AG).

**Figure 4 polymers-14-04695-f004:**
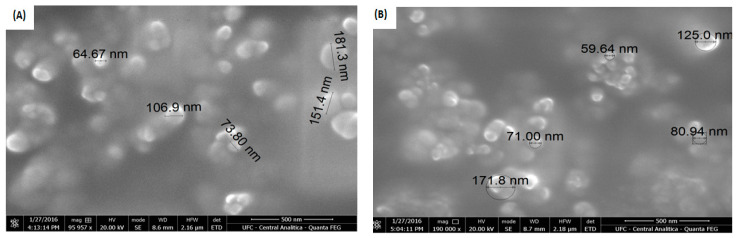
Micrographs of the (**A**) CHI/TPP/ALG and (**B**) CHI/TPP/AG NPs.

**Figure 5 polymers-14-04695-f005:**
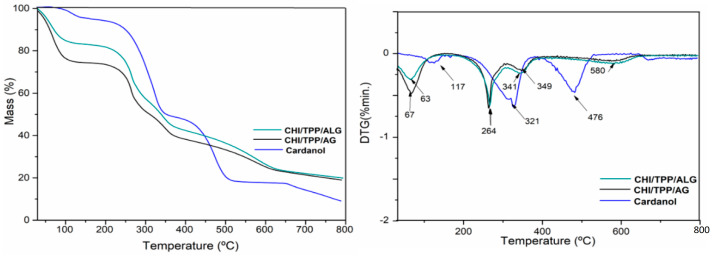
TGA curves for Cardanol and cardanol NPs CHI/TPP/ALG and CHI/TPP/AG; Mass (%) of the sample as a function of temperature, and first derivative (DTG) as a function of temperature.

**Figure 6 polymers-14-04695-f006:**
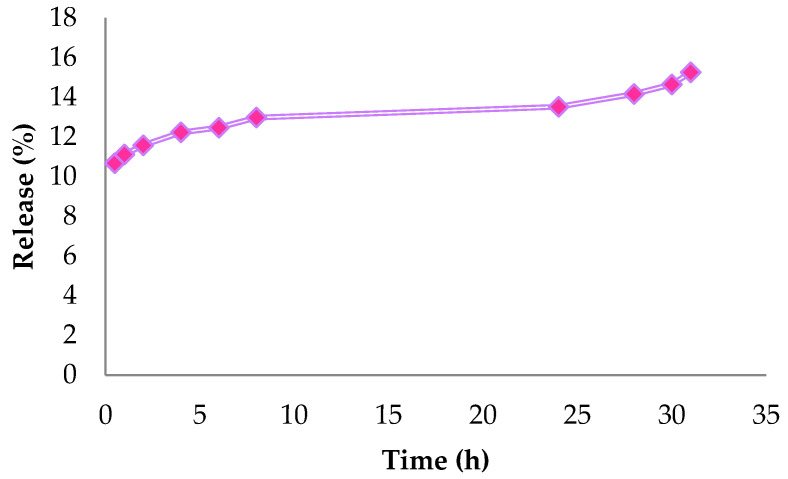
In vitro controlled release of CHI/TPP/ALG nanoparticles.

**Table 1 polymers-14-04695-t001:** Particle size and zeta potential evaluated as a function of the total volume fraction of NPs CHI/TPP/ALG and CHI/TPP/AG loaded with cardanol.

	Particle Size (nm)				Zeta Potential (mV)
NPs	PEAK 1		PEAK 2		
CHI/TPP/ALG	171.1	23.1%	34.9	76.9%	−39.7 ± 4.9
CHI/TPP/AG	312.2	52.2%	70.5	47.8%	−29.8 ± 4.0

**Table 2 polymers-14-04695-t002:** Weight loss and degradation temperature for cardanol, CHI/TPP/ALG, and CHI/TPP/AG nanoparticles.

Code NPs	Temperature Intervals (°C)	T(Max) (°C)	Weight Loss (%)	Residue at 700 °C
Cardanol	30–150	117.0	5.4	8.6
	150–375	321.3	46.8	
	375–800	475.8	39.8	
CHI/TPP/ALG	30–150	62.7	17.6	19.2
	150–307	264.3	27	
	307–450	341.0	16.5	
	450–800	580.0	19.7	
CHI/TPP/AG	30–150	66.7	25.6	19.0
	150–307	263.0	24	
	307–450	349.0	14.6	
	450–800	580	16.8	

## Data Availability

Not applicable.

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
