# Peer review of "Chitosan-Based Nanoparticles for Cardanol-Sustained Delivery System"

_polymers, 2022, doi:10.3390/polym14214695_

Round 1

Reviewer 1 Report

The manuscript entitled "Chitosan-based nanoparticles for the sustained delivery system of cardanol" is quite an interesting idea which would attract readers. The article is fine, but some of the following issues need to be resolved.

- Please be consistent; the phrase (physical-chemical) has been written in several different ways.

-Figure 3: The authors postulated that physical-chemical interactions and cross-links between the biopolymers in the encapsulated nanoparticles are what caused the band's intensity to rise by 3400 cm-1 and 1625 cm-1 for (CHI/TPP/ALG and CHI/TPP/AG). To determine whether the formulation process influences such physical-chemical interactions, I advise performing a comparable analysis on all components as physical mixtures.

- Table 1: Particle size and zeta potential of nanoparticles; it is unclear whether the results presented pertain to loaded or empty nanoparticles; please include both and indicate whether there was a difference owing to drug loading.

- Figure 6. TGA do cardanol e nanopartículas: I don't understand this and other phrases like it in the manuscript. The figure's ligand is also not clear.

- Thermal analysis: Could you please mention the physical form of Cardanol? It's not clear from the data on the nanoparticles whether they are loaded or not. This should be made clear and the results of both loaded and unloaded particles should be compared.

Author Response

Dear, We want to thank the reviewers for their crucial and constructive criticism, which allowed us to improve the scientific quality of our work immensely. Please find below the answers to their comments and questions. All modifications have been marked in yellow in the revised version of the manuscript to facilitate verification. Reviewer 1 Comments and Suggestions for Authors The manuscript entitled "Chitosan-based nanoparticles for the sustained delivery system of cardanol" is quite an interesting idea which would attract readers. The article is fine, but some of the following issues need to be resolved. - Please be consistent; the phrase (physical-chemical) has been written in several different ways. Answer: Thank you for your comments and contributions. All suggestions were incorporated into the new version of the manuscript. Changes are in the revised manuscript. -Figure 3: The authors postulated that physical-chemical interactions and cross-links between the biopolymers in the encapsulated nanoparticles are what caused the band's intensity to rise by 3400 cm-1 and 1625 cm-1 for (CHI/TPP/ALG and CHI/TPP/AG). To determine whether the formulation process influences such physical-chemical interactions, I advise performing a comparable analysis on all components as physical mixtures. Answer: Thank you for your comments and contributions. All suggestions were incorporated into the new version of the manuscript. Changes are in the revised manuscript. - Table 1: Particle size and zeta potential of nanoparticles; it is unclear whether the results presented pertain to loaded or empty nanoparticles; please include both and indicate whether there was a difference owing to drug loading. Answer: Thank you for your comments and contributions. All suggestions were incorporated into the new version of the manuscript. Changes are in the revised manuscript. - Figure 6. TGA do cardanol e nanopartículas: I don't understand this and other phrases like it in the manuscript. The figure's ligand is also not clear. Answer: Thank you for your comments and contributions. All suggestions were incorporated into the new version of the manuscript. Changes are in the revised manuscript. - Thermal analysis: Could you please mention the physical form of Cardanol? It's not clear from the data on the nanoparticles whether they are loaded or not. This should be made clear and the results of both loaded and unloaded particles should be compared. Answer: Thank you for your comments and contributions. All suggestions were incorporated into the new version of the manuscript. Changes are in the revised manuscript.

Reviewer 2 Report

Comments:
The manuscript “Chitosan-based nanoparticles for the sustained delivery system of cardanol” by Flavia Oliveira Monteiro da Silva Abreu and colleagues introduced cardanol encapsulated chitosan-based NPs with 70% of the biodistribution ranging from 34 to 300 nm. The NPs can maintain the activity of cardanol. However, the reviewer believes that additional points of clarifications could potentially be addressed to further strengthen the manuscript.

1.     The title states the chitosan-based nanoparticles can achieve sustained delivery of cardanol, but there was no cardanol release data vs time to prove the sustained delivery in this report. The author needs to provide the release data, otherwise, I would recommend changing the title of the report.

2.     In line 95 and 96, the line space is different, please correct.

3.    In line 156 and 157, the equation font is different from the main text. Please correct.

4.     Figure 1 and Table 2 caption are aligned both sides, but Figure 3 and Figure 6 are aligned middle. Please make them consistent.

5.     There is no Figure 5 in the report but has Figure 6 in the report. Please correct.

6.     In Figure 4 and Table 1, the author provides the SEM images of the NPs CHI/TPP/ALG, and CHI/TPP/AG and the saying the diameter is less than 200 nm, but the PDI is not included. It is recommended that the authors include the PDI data.

7.     In line 215, the author states the rough surface of the particles, I would recommend use Atomic Force Microscope (AFM) to measure the surface roughness of the NPs.

8.     In line 234, the author lists particles made from other authors in the literature. What is the conclusion/discussion you want to make to include this paragraph here?

9.     In line 247, the author says that “particles with zeta potential value > 30 or <= 30 mV prevent undesirable oscillation.” This sentence means that all the particles with whatever zeta potential can prevent undesirable oscillation. Please clarify it.

10.  In line 247, the author says that “the sample showed good stability in an aqueous solution.” Please clarify what aqueous solution.

11.  In line 248, the author states that “the NPs CHI/TPP/GA showed less stability due to the smaller number of acid groups.” Please provide the stability data here. Also please provide any experiments to show that smaller number of acid groups is the reason led to the unstable of NPs CHI/TPP/GA.

12.  In line 275, the table 2 caption “Weight loss e degradation temperature for cardanol, nanoparticulas CHI/TPP/ALG and CHI/TPP/AG”. Word spelling errors here, please correct it.

13.  In line 278 “between 30 and 150°C” and line 298 “at 264°C”. Need a space here. Please correct.  

14.  In line 322, the sentence “The production of NPs based on natural polymers loaded with cardnol was successfully produced”. Grammarly error, need to correct.

15.  In conclusion part, the line space is different from main text. Need to correct.  

Author Response

Dear,

We want to thank the reviewers for their crucial and constructive criticism, which allowed us to improve the scientific quality of our work immensely. Please find below the answers to their comments and questions. All modifications have been marked in yellow in the revised version of the manuscript to facilitate verification.

Revisor 2:

Comments and Suggestions for Authors

Comments:
The manuscript “Chitosan-based nanoparticles for the sustained delivery system of cardanol” by Flavia Oliveira Monteiro da Silva Abreu and colleagues introduced cardanol encapsulated chitosan-based NPs with 70% of the biodistribution ranging from 34 to 300 nm. The NPs can maintain the activity of cardanol. However, the reviewer believes that additional points of clarifications could potentially be addressed to further strengthen the manuscript.

  1. The title states the chitosan-based nanoparticles can achieve sustained delivery of cardanol, but there was no cardanol release data vs time to prove the sustained delivery in this report. The author needs to provide the release data, otherwise, I would recommend changing the title of the report.

Answer: Thank you for your comments and contributions. All suggestions were incorporated into the new version of the manuscript. Changes are in the revised manuscript.

  1. In line 95 and 96, the line space is different, please correct.

Answer: Thank you for your comments and contributions. All suggestions were incorporated into the new version of the manuscript. Changes are in the revised manuscript.

  1.   In line 156 and 157, the equation font is different from the main text. Please correct.

Answer: Thank you for your comments and contributions. All suggestions were incorporated into the new version of the manuscript. Changes are in the revised manuscript.

  1. Figure 1 and Table 2 caption are aligned both sides, but Figure 3 and Figure 6 are aligned middle. Please make them consistent.

Answer: Thank you for your comments and contributions. All suggestions were incorporated into the new version of the manuscript. Changes are in the revised manuscript.

  1. There is no Figure 5 in the report but has Figure 6 in the report. Please correct.

Answer: Thank you for your comments and contributions. All suggestions were incorporated into the new version of the manuscript. Changes are in the revised manuscript.

  1. In Figure 4 and Table 1, the author provides the SEM images of the NPs CHI/TPP/ALG, and CHI/TPP/AG and the saying the diameter is less than 200 nm, but the PDI is not included. It is recommended that the authors include the PDI data.

Answer: Thank you for your comments and contributions. However, due to lack of data, it was not possible to process the figures.

  1. In line 215, the author states the rough surface of the particles, I would recommend use Atomic Force Microscope (AFM) to measure the surface roughness of the NPs.

Answer: The requested analysis cannot be performed due to lack of resources/equipment. However, references were added in the text of other studies that also obtained rough surfaces using these polymers. Thank you for your comments and contributions.

  1. In line 234, the author lists particles made from other authors in the literature. What is the conclusion/discussion you want to make to include this paragraph here?

Answer: Corroborate with the results obtained, since several other studies work with the same polymers used in this study for the encapsulation of cardanol.

  1. In line 247, the author says that “particles with zeta potential value > 30 or <= 30 mV prevent undesirable oscillation.” This sentence means that all the particles with whatever zeta potential can prevent undesirable oscillation. Please clarify it.

Answer: New citations and references have been inserted for better understanding of the text. Thank you for your comments and contributions. All suggestions were incorporated into the new version of the manuscript. Changes are in the revised manuscript.

  1. In line 247, the author says that “the sample showed good stability in an aqueous solution.” Please clarify what aqueous solution.

Answer: New citations and references have been inserted for better understanding of the text. Thank you for your comments and contributions. All suggestions were incorporated into the new version of the manuscript. Changes are in the revised manuscript.

  1. In line 248, the author states that “the NPs CHI/TPP/GA showed less stability due to the smaller number of acid groups.” Please provide the stability data here. Also please provide any experiments to show that smaller number of acid groups is the reason led to the unstable of NPs CHI/TPP/GA.

Answer: The paragraph was removed and other references were added to discuss the results obtained. Thank you for your comments and contributions. All suggestions were incorporated into the new version of the manuscript. Changes are in the revised manuscript.

  1. In line 275, the table 2 caption “Weight loss e degradation temperature for cardanol, nanoparticulas CHI/TPP/ALG and CHI/TPP/AG”. Word spelling errors here, please correct it.

Answer: Thank you for your comments and contributions. All suggestions were incorporated into the new version of the manuscript. Changes are in the revised manuscript.

  1. In line 278 “between 30 and 150°C” and line 298 “at 264°C”. Need a space here. Please correct.  

Answer: Thank you for your comments and contributions. All suggestions were incorporated into the new version of the manuscript. Changes are in the revised manuscript.

  1. In line 322, the sentence “The production of NPs based on natural polymers loaded with cardnol was successfully produced”. Grammarly error, need to correct.

Answer: Thank you for your comments and contributions. All suggestions were incorporated into the new version of the manuscript. Changes are in the revised manuscript.

  1. In conclusion part, the line space is different from main text. Need to correct.  

Answer: Thank you for your comments and contributions. All suggestions were incorporated into the new version of the manuscript. Changes are in the revised manuscript.

Round 2

Reviewer 1 Report

Although the authors claimed to have addressed every issue previously brought up, I believe that there are still some unanswered questions.

-Figure 3: The authors postulated that physical-chemical interactions and cross-links between the biopolymers in the encapsulated nanoparticles are what caused the band's intensity to rise by 3400 cm-1 and 1625 cm-1 for (CHI/TPP/ALG and CHI/TPP/AG). To determine whether the formulation process influences such physical-chemical interactions, I advise performing a comparable analysis on all components as physical mixtures.

- Table 1: Particle size and zeta potential of nanoparticles; it is unclear whether the results presented pertain to loaded or empty nanoparticles; please include both and indicate whether there was a difference owing to drug loading.

- Thermal analysis: Could you please mention the physical form of Cardanol? It's not clear from the data on the nanoparticles whether they are loaded or not. This should be made clear and the results of both loaded and unloaded particles should be compared.

Author Response

Dear,

We want to thank the reviewers for their crucial and constructive criticism, which allowed us to improve the scientific quality of our work immensely. Please find below the answers to their comments and questions. All modifications have been marked in yellow in the revised version of the manuscript to facilitate verification.

Comments and Suggestions for Authors

Although the authors claimed to have addressed every issue previously brought up, I believe that there are still some unanswered questions.

 -Figure 3: The authors postulated that physical-chemical interactions and cross-links between the biopolymers in the encapsulated nanoparticles are what caused the band's intensity to rise by 3400 cm-1 and 1625 cm-1 for (CHI/TPP/ALG and CHI/TPP/AG). To determine whether the formulation process influences such physical-chemical interactions, I advise performing a comparable analysis on all components as physical mixtures.

Answer: Thank you for your comments and contributions. All suggestions were incorporated into the new version of the manuscript. Changes are in the revised manuscript.

- Table 1: Particle size and zeta potential of nanoparticles; it is unclear whether the results presented pertain to loaded or empty nanoparticles; please include both and indicate whether there was a difference owing to drug loading.

Answer: Thank you for your comments and contributions. All suggestions were incorporated into the new version of the manuscript. Changes are in the revised manuscript.

- Thermal analysis: Could you please mention the physical form of Cardanol? It's not clear from the data on the nanoparticles whether they are loaded or not. This should be made clear and the results of both loaded and unloaded particles should be compared.

Answer: Thank you for your comments and contributions. All suggestions were incorporated into the new version of the manuscript. Changes are in the revised manuscript.

However, thermal analysis was performed only for NPs loaded with cardanol, in order to protect the active ingredient.

Reviewer 2 Report

 Overall the authors addressed most of my comments, but there are still some small errors need to be corrected. 

1.     In line 95 a heating rate of 10 °C.min-1, this need to be corrected.

2.     In Figure 3, the x-axis and y-axis fond size and shape looks weird, please correct

3.     From line 228-230, the font size is different from the main text, please correct.

Author Response

Dear,

We want to thank the reviewers for their crucial and constructive criticism, which allowed us to improve the scientific quality of our work immensely. Please find below the answers to their comments and questions. All modifications have been marked in yellow in the revised version of the manuscript to facilitate verification.

Comments and Suggestions for Authors

 Overall the authors addressed most of my comments, but there are still some small errors need to be corrected.

  • In line 95 a heating rate of 10 °C.min-1, this need to be corrected.

Answer: Thank you for your comments and contributions. All suggestions were incorporated into the new version of the manuscript. Changes are in the revised manuscript.

  • In Figure 3, the x-axis and y-axis fond size and shape looks weird, please correct.

Answer: Thank you for your comments and contributions. All suggestions were incorporated into the new version of the manuscript. Changes are in the revised manuscript.

  • From line 228-230, the font size is different from the main text, please correct.

Answer: Thank you for your comments and contributions. All suggestions were incorporated into the new version of the manuscript. Changes are in the revised manuscript.

Kind regards.
